# Pleuropneumonectomy as Salvage Therapy in Children Suffering from Primary or Metastatic Sarcomas with Pleural Localizations

**DOI:** 10.3390/cancers13153655

**Published:** 2021-07-21

**Authors:** Frédéric Hameury, Perrine Marec-Berard, Mathilde Eymery, Marc H. W. Wijnen, Niels van der Kaaij, Pierre-Yves Mure, François Tronc, Franck Chotel, Clara Libbrecht, Wim Jan P. van Boven, Lianne M. Haveman

**Affiliations:** 1Department of Pediatric Surgery, Hôpital Femme Mère Enfant, Hospices Civils de Lyon, Claude Bernard University, 69677 Bron, France; frederic.hameury@chu-lyon.fr (F.H.); pierre-yves.mure@chu-lyon.fr (P.-Y.M.); francois.tronc@chu-lyon.fr (F.T.); franck.chotel@chu-lyon.fr (F.C.); 2Institute of Hematology and Pediatric Oncology, 69008 Lyon, France; perrine.marec-berard@ihope.fr (P.M.-B.); mathilde.eymery@chu-lyon.fr (M.E.); clara.libbrecht@childrens.harvard.edu (C.L.); 3Princess Maxima Center for Pediatric Oncology, 3584 CS Utrecht, The Netherlands; M.H.W.Wijnen-5@prinsesmaximacentrum.nl; 4Department of Cardiothoracic Surgery, University Medical Center Utrecht, 3584 CX Utrecht, The Netherlands; N.P.vanderKaaij-2@umcutrecht.nl; 5Department of Cardiothoracic Surgery, Amsterdam University Center, Location AMC, 1105 AZ Amsterdam, The Netherlands; w.j.vanboven@amsterdamumc.nl

**Keywords:** pleuropneumonectomy, pediatric, sarcoma, quality of life, outcome

## Abstract

**Simple Summary:**

Pediatric sarcoma patients with pleuropulmonary lesions have a dismal prognosis because the impossibility to achieve local control. Local therapy with radiotherapy, whether in combination with chemotherapy, appears to be insufficient to eliminate the tumor cells. The aim of this study was to determine if pleuropneumonectomy (PP) could be a therapeutic option. We retrospectively reviewed nine patients who underwent PP for pleuropulmonary localization of primary localized sarcoma or metastatic recurrence. Surgery and complications were analyzed, pulmonary function tests were conducted, and quality of life was determined. Outcome is variable, four patients died within 14 months after PP, one patient suffered from local recurrence, and four patients are in long-lasting remission. This extended operation is quite well tolerated. Lung function seems preserved, and quality of life is generally good. Because it gives improvement of survival in patients with pleural lesions, PP can be considered as effective salvage therapy in selected patients.

**Abstract:**

Pediatric sarcoma patients with pleuropulmonary lesions have a dismal prognosis because the impossibility to obtain local control. The aim of this study was to determine if pleuropneumonectomy (PP) could be a therapeutic option. We retrospectively reviewed nine patients who underwent salvage PP for pleuropulmonary localization of primary localized sarcoma or metastatic recurrence. Surgery and complications were analyzed, pulmonary function tests were conducted, and quality of life was determined with EORTC-QLQ-C30 questionnaire. At the time of PP age was between 9–17 years. Underlying disease included metastatic osteosarcoma (*n* = 5), Ewing sarcoma (two metastatic, one primary), and one primary undifferentiated sarcoma. Early complications occurred in three patients. Mean postoperative hospitalization stay was 14.5 days. Pulmonary function test showed 19–66% reduction of total lung capacity which led to mild exercise intolerance but did not affect daily life. Four patients died of multi-metastatic relapse <14 months after PP, one patient had a local recurrence, and four patients are in complete remission between 1.5 and 12 years after PP. In conclusion, in this small patient group treated with a pleuropneumonectomy for primary or metastatic lesions, outcome is variable; however, this extended surgical technique was generally quite well tolerated. Postoperative lung function seems well preserved, and it seems to lead to at least an extension of life with good quality and therefor can be considered as salvage therapy.

## 1. Introduction

Pediatric sarcoma patients with pleuropulmonary lesions, either initial or found at a later stage, have a dismal prognosis due to the lack of local control [1,2]. Radiotherapy, whether combined with chemotherapy, appears to be insufficient to eliminate the malignant tumor cells and to prevent recurrence or progression of disease [1,2]. For intrapulmonary lesions, metastasectomy is an option; however, for pleural lesions, a radical or pleuropneumonectomy (PP) seems the only curative option to obtain local control by complete resection. PP is a standardized procedure with en bloc resection of the parietal and visceral pleura of the lung, the lung, pericardium, and homolateral diaphragm [3]. This procedure was first used in adult patients with tuberculosis infection [4] and subsequently applied in adult patients with malignant pleural mesothelioma [5,6,7] and thymomas [8,9], with mainly loco regional involvement of disease. This procedure is also described in patients with lung metastases [10], and primary sarcomas of the hemithorax [11,12,13]. In this procedure about one-third of all adult patients experienced a surgical complication after PP and post-operative mortality rates up to 10 percent are reported [14,15]. In children, pneumonectomy is especially performed for benign indications such as post-infectious bronchiectasis or pulmonary malformations [16,17,18,19,20]. Extrapleural pneumonectomy or pleuropneumonectomy for pediatric malignancies involving the pleura is described in very few articles and little is known about the indication and outcome for pediatric malignancies [16,21,22]. Primary and metastatic malignant thoracic tumors in children frequently concerns sarcoma, especially Ewing sarcoma, osteosarcoma, or synovial sarcoma [1,2,23].

In general practice, most of the time PP is only accepted for patients with primary localized intrathoracic sarcoma. In case of recurrence of a thoracic relapse, given the poor prognosis, only surgery with limited pleural resection is often performed, only lung sparing surgery with limited pleural resection is usually proposed. Little is known about morbidity and mortality, however this seems maybe better compared with adults [16,17,18,19,20,21,22]. The aim of this study is to determine whether PP is a treatment option for patients with primary, and also recurrence of metastatic pleuropulmonary sarcoma. To this end, we report our experience with morbidity, but with a particular focus on quality of life and long-term outcome.

## 2. Materials and Methods

### 2.1. Study Population

Data from patients who underwent a salvage PP for primary or secondary pleuropulmonary location of sarcoma between 2005 and 2020 were retrospectively reviewed. Patients were treated in the department of pediatric hematology and oncology (IHOP, France), in the pediatric surgery department of the university medical center of Lyon (Hospices Civils de Lyon), France, and in the Academic Medical Center Amsterdam, the Netherlands or in the Princess Maxima Center for Pediatric Oncology/University Medical Center Utrecht, the Netherlands. For Dutch patient’s, PP were the result of a collaboration between Dutch and French surgery teams.

Patient records were analyzed to collect clinical information including diagnosis, initial tumor localization, chemotherapy, feature and time of recurrence, surgery, early and late complications, length of stay in hospital, histological response (HR) (good HR ≤ 10% vital tumor cells; poor HR ≥ 10% vital tumor cells), addition of radiotherapy, and time-dependent survival (Event-Free survival (EFS) and Overall Survival (OS)). EFS time was defined as the interval between the date of PP and the date of first event or death. Event was defined as progressive disease, relapse of disease (local or metastatic), secondary malignancy, or death. Patients were censored at the date of most recent consultation. OS was defined as the time from PP until death from any cause. Surviving patients were evaluated at the date of last contact.

Pulmonary Function Tests (PFT) were performed before and after surgery. Pulmonary function was tested by a combining body plethysmography and spirometry and is validated for children above 6 years. The Total Lung Capacity (TLC) was seen as the main criterion, as it is used to define restrictive lung conditions, representing the main consequence which can be expected after total lung resection. Forced Expiratory Volume in one second (FEV1) and Forced Vital Capacity (FVC) were also used to monitor the function over the time.

Quality of life was evaluated with the European Organization for Research and Treatment of Cancer Quality of Life Questionnaire (EORTC-QLQ-C30) [24]. This is a cancer specific questionnaire with 30 items. Three scores were calculated: the Functional Scale score (5 domains: physical, emotional, social, cognitive, and role-playing), the Symptom Scale score (3 domains: pain, fatigue and nausea, and vomiting), and a score on a 2-item Global Health Scale. All questions were answered on a 4-level Likert-type scale, except the global health score which has a 7-level scale. All items are linearly transformed into scales ranging from 0 to 100. Higher scores on functional and Global Health Score indicate a higher level of functioning and high quality of life, while a higher score on the Symptom Scale indicates that patients experience more discomfort in daily life. The Lansky score was used as described earlier [25].

Data were collected with approval of Institutional Review Boards and informed consent for data collection was obtained from all patients.

### 2.2. Treatment

All patients diagnosed with a sarcoma started with treatment according to the different sarcoma treatment protocols (see Table 1). In all these protocols, treatment starts with induction chemotherapy followed by local therapy (surgery and/or radiotherapy) and subsequently maintenance chemotherapy. In case of recurrence of disease, different treatment options were given to the patients as shown in Table 1.

### 2.3. Indication for PP

The indication for PP as salvage local therapy was discussed in a multidisciplinary tumor board. Patients were eligible for PP if they met the following criteria:-The only alternative to PP was palliative treatment.-The pleural lesions were limited to one hemithorax, possibly associated with one or more lesions within the ipsilateral lung parenchyma. No other metastatic localizations were found.-Patients received preoperative chemotherapy with sufficient response.-The patient must be in a reasonable good general condition, i.e., a Lansky performance scale above 80 (i.e., active but tires more quickly) [25].

Beside the pulmonary lung function test a ventilation–perfusion scan and cardiac evaluation were performed as part of the standard work-up before PP in all patients.

In Figure 1, imaging of two patients with a pleuropulmonary localization of a primary localized osteosarcoma ((1a), patient nr.3, Table 1)) and with a pleural relapse of an osteosarcoma ((1b), patient nr. 9, Table 1)) are shown. The indications for radiotherapy were based on positive resection margins, lymphatic emboli, type of sarcoma, or poor histological response.

### 2.4. Surgery

Surgical steps were adjusted according to the location of the tumor as shown on the preoperative CT scan. A lateral thoracotomy was performed either by an extended lateral incision through two intercostal spaces or by a partial longitudinal sternotomy with anterolateral thoracotomy (Hemiclamshell). Pleural adhesions due to previous surgery need to be removed to enable access and ribs invaded by the tumor lesion had to be removed. This could lead to parietal reconstruction at the end of the procedure. In the absence of a distinct plane of dissection between the mediastinal pleura and the pericardium—which is more often the case when the child is older, a resection of the pericardium was necessary, followed by its reconstruction. The same difficulties in separating the parietal pleura from the diaphragm most often led to limited resection of the diaphragm. If this resection was extensive, due to tumor invasion for example, a Gore-Tex patch was used. Depending on the central extent of the tumor, the hilar vessels had to be transected intra- or extrapericardially. The specimen including pleura, lung and diaphragm, with or without pericardium, would be removed en bloc. Pleural breaches were common, but at distance from the tumor location which made the pleura thick and solid. The location of uncertain tumor spreading or margins were identified with metal clips to help following radiotherapy.

### 2.5. Radiotherapy

Both in the primary treatment as well in the situation of recurrence of disease radiotherapy was given according to the treatment protocols used in the treatment of Ewing sarcoma (Ewing 1999 and Ewing 2008 protocol) and osteosarcoma (Euramos protocol). Ewing sarcoma are radiosensitive tumors. Surgery is favored whenever feasible and definitive radiotherapy is only used for inoperable tumors, or in combination with surgery, also special localizations could be treated with radiotherapy alone [26]. Postoperative radiotherapy is indicated in intralesional or marginal surgery and in poor histological response regardless of surgical margins. In osteosarcoma, complete surgery is the local therapy of choice. Radiotherapy is only added in situations where complete surgery cannot be achieved and is recommended for inoperable sites or those that could only be operated with inadequate margins.

## 3. Results

### 3.1. Patient Characteristics

Nine patients with primary sarcoma or metastatic recurrence underwent PP as salvage therapy. The characteristics of these patients are reported in Table 1. Five male and four female patients had a diagnosis of sarcoma at a median age of 15 years (range between 6 and 16 years). At the time of PP, patient’s age ranged from 9 to 17 years. Underlying disease was osteosarcoma in five, Ewing sarcoma in three, and undifferentiated chest wall sarcoma in one. Pleural lesions were present at diagnosis in three patients (two Ewing sarcoma, and one undifferentiated sarcoma) or at relapse in six patients (five osteosarcoma and 1 Ewing sarcoma). In this last group, the pleuropulmonary lesions appeared 11 to 31 months (mean 22.6 months) after the end of the first line treatment (Table 1).

### 3.2. Pre- and Postoperative Treatment

All patients received neoadjuvant chemotherapy with a good radiological response. PP was performed during second line treatment in eight cases and during the third line in one (patient nr. 6). All adjuvant chemotherapy could be started within 21 days of surgery. Postoperative radiotherapy, either or not combined with chemotherapy, was given in order to obtain complete local control and was given because of positive resection margins, positive lymph nodes or poor histological response as decided in a multidisciplinary tumor board. This was the case in six patients (Table 1, pat.nr. 2–7; 3 Ewing sarcoma, 2 osteosarcoma and 1 undifferentiated sarcoma). In five patients, a poor histological response was seen. In four of the five patients, indeed radiotherapy was added conform treatment protocols. Only in one patient with an osteosarcoma (pat.nr. 1) no radiotherapy was added because complete surgery was achieved.

### 3.3. Surgical Treatment

In Table 2, modalities of the PP procedure in individual patients are detailed. Lateral thoracotomy through two intercostal spaces in eight patients and a hemiclamshell procedure in one patient allowed six left and three right pleuropulmonary resections. The diaphragm was partially (4/9) or totally (5/9) resected. Pericardium was also reconstructed after resection in two patients with a Gore-Tex patch. The medium arch of the third and fourth rib was removed in one case. Thoracotomies and sternotomies allowed complete resection in 3/9 and marginal resection in 6/9 patients.

Acute perioperative complications did not occur in any of the patients. Early post-operative complications occurred in three patients. One patient (Table 1, pat.nr. 8) had a pulmonary infection possibly combined with infection of the Gore-Tex patch of his diaphragm that required long-term antibiotic treatment. Postoperative bleeding requiring surgery occurred in one patient (Table 1, pat.nr. 6), and one patient developed a cardiac tamponade, necessitating an emergent pericardial window (Table 1, pat.nr.9). The last patient was also treated with antibiotics because of a pneumonia. The mean postoperative hospitalization stay was 14.5 days (range 8 to 24).

Late postoperative complications occurred in two patients: one presented with an ulcerated esophagitis due to a hiatal hernia and was treated by surgical repair, and the other developed feeding problems without evidence of hiatal hernia or gastroesophageal reflux disease, however resolved after gastrostomy.

### 3.4. Pulmonary Function Tests

Preoperative PFT were performed before PP in all but one patient (Table 2) and showed a reduced pulmonary function compared to the values expected for children of the same age and sex range, mean). Post-operative PFT were performed in eight patients and showed a 19 to 66% reduction in TLC. The highest reduction was seen in a patient who developed a severe scoliosis, which required surgery with additional treatment with a corset. The main symptom related to the reduction in TLC was dyspnea with mild physical exercise, but this did not affect daily life in most of the children.

### 3.5. Survival and Recurrence

Mean follow-up time is 6.8 years. Five of the nine patients are still alive. Four patients died because of a multi-metastatic relapse. Two patients had an early recurrence 2 and 3 months after PP, respectively, and two patients had later recurrence of disease, respectively, 10 and 14 months after PP. One patient (patient nr. 9) suffered from a local recurrence in the former right hilus region 7 months after PP and started again with chemotherapeutic treatment. Four patients are still alive with complete remission, respectively, 1.5, 7, 11, and 14 years after PP. Three patients with metastatic recurrence of an osteosarcoma (Table 1, pat.nr.1, 2, 8) and one patient with pleural involvement of an Ewing sarcoma at diagnosis (Table 1, pat.nr. 3). Four of the five osteosarcoma patients alive were treated for a solitary pleural lesion at time of recurrence. In one patient (pat.nr.9, Table 1), one lesion was seen at the CT and was removed thorocoscopically; however, at the time of surgery, also three other small pleural lesions were detected. Tumor volume at time of recurrence was different, the largest tumor measured 28 cm × 6 cm × 12 cm and the smallest lesion measured 3 cm × 2.7 cm × 3 cm. Before PP there was either a good radiological response, or in case the size remains the same a biopsy prior PP was taken.

### 3.6. Quality of Life

Six months after the TPP, all seven surviving patients had a Lansky score above 80%. All children could go back to school or go back to daily life shortly (mostly within four weeks), after surgery. One patient (pat.nr.8) went to a rehabilitation center 4 months after the PP, just before he had a pleural recurrence of his osteosarcoma. Five surviving patients completed the Quality of life questionnaire. The symptom scores were under 20% and the Functional and Global Health Scale was greater than 67% for all but one patient. The two patients who died more than 10 months after surgery (pat.nr. 5 and 6) and the patient with recurrence after PP (pat.nr.9) reported that they did not regret the PP. The latter stated that he has a good quality of life and knows that everything has been done to cure him.

## 4. Discussion

Pleuropulmonary pediatric sarcoma, either primitive or metastatic, has a very poor prognosis, particularly in case of recurrence. The overall survival at 1 year is close to zero [1,2]. Polites et al. reported no survivor at one year after pneumonectomy for six metastatic osteosarcoma patients, suggesting pneumonectomy was not a good indication for these patients [22]. In our study, 9 months after PP, two patients died early of multi-metastatic relapse, and two died more than 10 months after PP. One patient suffered from a local recurrence 7 months after the PP and has restarted chemotherapy with initial response. Four patients are in complete remission between 1.5 and 14 years after PP. Four of the patients alive were treated for pleuropulmonary recurrence of osteosarcoma, three of these patients had a solitary lesion, and in one patient three other pleural lesions were detected at time of biopsy. One patient was treated for a primary Ewing sarcoma with pleural involvement. Therefore, PP seems to significantly improve overall survival in a selected group of patients even in the case of a relapse. Numerous conditions must be met. First of all, PP should be the only curative option to obtain local control. The lesions have to be limited to one hemi-thoracic side and there must be no other metastatic lesions other than the pleuropulmonary lesions. Second, we think a good radiological response to chemotherapy is essential because of the importance to treat also microscopic disease, or if the size remains the same, demonstrating that the tumor has responded to chemotherapy by means of a biopsy. In pediatric bone sarcoma a good radiological response is correlated with better outcome, although a direct correlation with the histological response has not been demonstrated [27,28]. However, it is well known that chemotherapy induced necrosis is a predictive factor for prognosis [29,30,31]. In five out of the nine patients, a poor histological response after PP, conform treatment protocols we think that radiotherapy should be added, except in patients with osteosarcoma where complete surgery could be achieved. Based on our experience, PP should not be offered to patients with progression under therapy. A disease-free interval of more than 1 year correlates with better survival in patients with osteosarcoma [32], thus patients with late recurrence of the disease may benefit more from PP. Nevertheless, the overall survival is still low, and clear counseling regarding the low chance of survival after this surgery is essential concerning the risk of complications.

Despite better morbidity and mortality rates in children than in adults after pneumonectomy for benign indications [16,17,18,19,20], PP is barely used for pediatric malignancies and sarcoma. The reason is probably the bad reputation of PP in term of complication and morbidity. Recently, Polite et al. described 38 pneumonectomy for malignancies (mainly osteosarcoma, inflammatory myofibroblastic tumors, and pleuropulmonary blastoma) in pediatric patients over 34 institutions. The early complication rate was 25 percent (10 out of 38 patients), including one death [22]. In adults, up to half of patients experienced early complications after PP and mortality rates up to 10 percent [10,14,15]. Arrhythmias are common in adults (40%), but rare in children with only one case described [21]. Other complications frequently described in adults, and which can lead to significant mortality, are pulmonary edema, bronchopleural or esophagopleural fistula, chylothorax, and cardiac herniation. The post-pneumectomy syndrome may occur 6 months to 35 years after the procedure [33,34]. To prevent this complication in young children, prophylactic expander insertion appears to be a safe option [33]. Reported long-term complications, i.e., scoliosis (in children), esophageal motility disorders, chronic chest pain, and vocal cord dysfunction may occur months to years after PP [17,19]. Early postoperative complications occurred in three patients, including infection, postoperative bleeding, and cardiac tamponade, but none of the complications as described above. The mean postoperative hospitalization stay was two weeks (range: 8–24 days). Late complications related to surgery were feeding difficulties and hiatal herniation. One severe scoliosis needed a surgery, which is a common complication in extended surgery because of thoracic sarcoma [35,36]. Close follow-up is important in order to make early correction if needed. To explain this low rate of severe complications several arguments may be advocated and the main one is the cooperation between adult thoracic surgeons and pediatric surgeons, both sharing their experiences. In a second thought it seems the younger the child the easier the PP is, whereby diaphragm and pericardium sometimes can be preserved.

Almost all patients suffered from exercise related dyspnea. Although the evaluation of postoperative pulmonary function and quality of life has been influenced by survival bias, in our patients the post-operative pulmonary function decreased by less than fifty percent as could be expected by resection of a complete lung. This could be explained by the fact that there is overexpansion of the remaining lung. This is in keeping with previous studies that showed that the respiratory function is well preserved in children who underwent pneumonectomy [22,37]. One of the major arguments against PP is the fear of impaired quality of life in survivors but even more so in those who will die shortly after the intervention, the fear of over-treatment. However, postoperative respiratory function and exercise-related dyspnea did not seem to be, or were hardly affected by, daily life activities. Patients went back to school or restarted with rehabilitation shortly after the intervention. Four of the five patients who completed the QoL questionnaire showed a good function with less symptoms. Furthermore, patients with recurrence of disease did not regret that they underwent the PP as it has given extension of life with good quality.

Although our patient group is the largest study concerning PP for pediatric sarcoma patients as far as we know, the sample size is very low, according to the rarity of patients eligible for this procedure. Further follow-up is necessary. The long-term results regarding morbidity and mortality need to be confirmed in a larger, multicenter prospective study.

## 5. Conclusions

Although outcome is variable in patients treated with a pleuropneumonectomy for primary or metastatic pleural lesions, this extended surgical technique is generally quite well tolerated. Postoperative lung function seems well preserved, and in almost all patients, quality of life is good. To our knowledge, no other treatment demonstrated such improvement of survival, even if it must be confirmed with largest studies. Therefore, PP seems to be an alternative to palliative treatment in a selected group of young patients with pleural sarcoma.

## Figures and Tables

**Figure 1 cancers-13-03655-f001:**
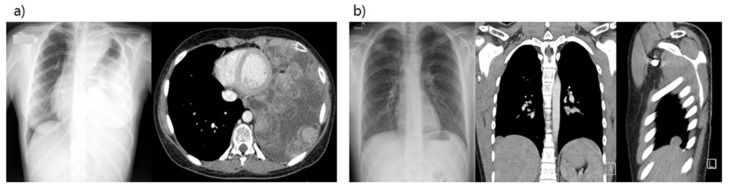
(**a**) Imaging of a patient with a pleuropulmonary localization of a primary localized Ewing sarcoma of the chest wall and (**b**) imaging of a patient with a pleural relapse in the left lateral sinus of an osteosarcoma.

**Table 1 cancers-13-03655-t001:** Patient characteristics who underwent total pleuropneumonectomy (PP) for pleuropulmonary localization of a sarcoma and non-surgical treatment. M = Methotrexate, E = etoposide, I = ifosfamide, A = adriamycin, P = cisplatin, G = gemcitabin, D = docetaxel, Tem = temozolomide, Iri = irinotecan, C = cyclophosphamide, Ca = carboplatin, RT = radiotherapy, HD-CT high-dose chemotherapy, aSCT = autologous stem cell transplantation, BuMel = busulphan, melphalan, EFS = event-free survival, OS = overall survival.

Pat.nr.	Age at Diagnosis/Sex	Initial Diagnosis	First Line Chemotherapy According Treatment Protocol	Time between End of Treatment and Relapse with Pleuropulmonary Lesions	Number of Relapses	2nd Line Chemo-Therapy	Histological Response after PP	Additional Therapy after PP	EFS (Months/Years)	OS (Months/Years)	Patient Status (Months/Years after PP)
1	15 y/Male	Osteosarcoma of the femur	French OS2006 protocol M-EI courses	28 months	1	IE–APx4	Poor > 10% viable cells	No	7.1 years	7.1 years	Alive
2	6 y/Male	Osteosarcoma of the humerus	French OS2006 protocol M-EI courses	11 months	1	APx2	Poor > 50% vital tumor cells.	RT and HD-CT with Thiotepa followed by aSCT	11.3 years	11.3 years	Alive
3	15 y/Female	Ewing sarcoma of the chest wall	Ewing 99 protocol	Pleural involvement at diagnosis	-	no	Good, rare viable cells.in pleura infiltrated	Hemithoracic RT with boost and VAI x7	14 years	14 years	Alive
4	10 y/Female	undifferentiated sarcoma of the chest wall	IA 4x + ICE courses	Pleural involvement at diagnosis	-	AP and palliative vinblastin	Poor > 50% viable cells	Hemithoracic RT	2 months	3 months	Died
5	10 y/Female	Ewing sarcoma of the pelvis with lung mets	Ewing 99 protocol VIDE courses followed by aSCT (BuMel)	29 months	2 all in left lung	TemIri	Poor > 50% viable cells	Hemithoracic RT and Tem + CPT11 vinorelbin + C	7 months	10 months	Died
6	13 y/Female	Osteosarcoma of the femur	Euramos protocol: MAP courses + IE because of radiological progression	22 months	3	GD	Poor vital tumor cells in pleura parietalis	RT and Ca E courses	10 months	13 months	Died
7	16 y/Male	Ewing sarcoma of the rib	Ewing 2008 protocol	Pleural involvement at diagnosis, early recurrence with pleural lesions after primary surgery	1	TemIri	Good, no vital tumor cells, but vital cells in biopsy of diaphragm	GD + RT	1.5 months	5 months	Died
8	15 y/Male	Osteosarcoma of the femur	Euramos protocol: MAP courses	31 months	1	IE	Good < 5% vital tumor cells, no infiltration in pleura	no	1.5 years	1.5 years	Alive
9	15 y/Male	Osteosarcoma of the femur	Euramos protocol: MAP courses	15 months	1	IE	Good, 0% tumor cells, no infiltration in pleura	GD after local recurrence	7 months	11 months	Alive, with recurrence of disease

**Table 2 cancers-13-03655-t002:** Information regarding the total pleuropneumonectomy (PP), complications, hospitalization days, lung function and scores on the quality of life questionnaire of individual patients ICS= intercostal space, TLC: total lung capacity, FEV1: forced expiratory volume in 1 s.

Pat.nr.	Technique + Resection Margins	Early Complications (<4 Weeks)	Days in Hospital after PP	Pulmonary Function Test		Late Complications (>4 Weeks)	Scores on Quality of Life Questionnaire
				Pre- Operative	Post- Operative		
1	-PP right side -Lateral thoracotomy (7th ICS) -Complete resection of diaphragm -Limits in sano, marginal resection	none	16	TLC: 68% FEV1: 80%	4.5 years after PP TLC: 50% FEV1: 37%	Mild dyspnea on exertion	5 years after PP Functional Score: 33% Global Health Score: 44.6% Symptom Score: 67%
2	-PP left side -Lateral thoracotomy (5th and subsequently 9th ICS) -Resection of the medial part of the diaphragm -Limits in sano, marginal resection -No infiltration in pleura	None	8	TLC: 104% FEV1: 111%	3 years after PP TLC: 53% FEV1: 48% 6 years after PP TLC: 38% FEV1: 38%	Severe scoliosis (Cobb angle >25°) Mild dyspnea on exertion	6 years after PP Functional Score: 98% Global Health Score: 91.6% Symptom Scales: 0%
3	-PP left side -Lateral thoracotomy (5th then 9th ICS) -Resection part of diaphragm -Partial pericardium resection -Limits not in sano, complete resection	None	15		5.5 years after PP TLC: 66% FEV1: 83.5% 8 years after PP TLC: 81% FEV1: 53%	Hiatal herniation with ulcerated esophagitis Asymmetrical breasts Mild dyspnea on exertion	9 years after PP Functional Score: 85% Global Health Score: 75% Symptom Score: 20%
4	-PP left side -Hemiclamshell -Resection of the medium arch of the 3rd and 4th rib -Positive resection margins	None	8	TLC: 71% FEV1: 77%	1 month after PP TLC: 49% FEV1: 52.5%		
5	-PP left side -Lateral thoracotomy (5th then 9th ICS) -Resection of major medial part of diaphragm -Positive resection margins	None	10	TLC: 67% FEV1: -	Not performed due to early progression	Feeding difficulties resolved after gastrostomy	
6.	-PP left side -Posterolateral thoracotomy (6th ICS, subsequently 1st ICS) -Complete resection hemidiaphragm -Part of pericard: reconstruction with patch -positive resection margins	Hematothorax due to re-bleeding → re-surgery 4 days after PP	17	TLC: 67% FEV1: 62% (after earlier lobectomy)	9 months after PP TLC: 44% FEV1: 41.5%	Mild dyspnea complaints in case of moderate physical activity	
7.	-PP right side -Posterolateral thoracotomy -Complete resection hemidiaphragm -Pericard patch -positive resection margins	None	17	TLC: 71% FEV1: 78%	Not performed due to early progression		
8.	-PP left side -Posterolateral thoracotomy (4th ICS) -Complete resection hemi -diaphragm -complete resection	Infection (day 7), requiring long term antibiotics	16	TLC: 85% FEV1:78%	8 months after PP TLC: 50% FEV1: 42%	Mild dyspnea complaints in case of moderate physical activity	18 months after PP Functional Score: 90% Global Health Score: 67% Symptom Score: 6%
9.	-PP right side -Posterolateral thoracotomy -Complete resection hemidiaphragm -Resection of small part pericardium -complete resection	Pneumonia (day 3) pneumopericardium (day 3) -cardiac tamponade → re-surgery: pericardial window (day 7)	24	TLC: 72% FEV1: 67% (earlier wedge excision)	9 months after PP TLC: 38% FEV1: 41%	Slightly reduced cardiac function 5 months after PP → ACE-inhibitor (same range as function before PP) Dyspnea in case of physical activity	9 months after PP Functional Score: 77% Global Health Score: 75% Symptom Score: 3%

## Data Availability

Informed consent was obtained from all subjects involved in the study.

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
