# Peer review of "Pleuropneumonectomy as Salvage Therapy in Children Suffering from Primary or Metastatic Sarcomas with Pleural Localizations"

_cancers, 2021, doi:10.3390/cancers13153655_

Round 1

Reviewer 1 Report

File attached

Author Response

We thank the reviewer for the comments. We will try to answer the comments point-by-point.

  • In the Results section in Table 1, we show the histological response of the 9 patients who underwent. In total five patients show a poor histological response and four patients had a good histological response after earlier chemotherapeutic treatment. The question of the reviewer is whether the results of the histological response support the conclusion of the study.

The reviewer is right, we didn’t write anything about the histological response and its relationship with outcome. Therefore we added some sentences in the Results section:

‘In 5 patients a poor histological response was seen. In four of the five patients indeed radiotherapy was added conform treatment protocols. Only in one patient with an osteosarcoma (patnr 1) no radiotherapy was added because complete surgery was achieved. (line 177-179)’

To clarify the indication of post-operative radiotherapy we also added a heading in the Method section concerning the indication of radiotherapy:

Radiotherapy

Both in the primary treatment as well in the situation of recurrence of disease radiotherapy was given according the treatment protocols used in the treatment of Ewing sarcoma (Ewing 1999 and Ewing 2008 protocol) and osteosarcoma (Euramos protocol). Ewing sarcoma are radiosensitive tumors. Surgery is favored whenever feasible and definitive radiotherapy is only used for inoperable tumors, or in combination with surgery, also special localization could be treated with radiotherapy alone [26]. Postoperative radiotherapy is indicated in intralesional or marginal surgery and in poor histological response regardless of surgical margins. In osteosarcoma complete surgery is the local therapy of choice. Radiotherapy is only added in situations where complete surgery cannot be achieved and is recommended for inoperable sites or those that could only be operated with inadequate margins.

26  Andreou D, Ranft A, Gosheger G, Timmermann B, Ladenstein R, Hartmann W, Bauer S, Baumhoer D, van den Berg H, Dijkstra PDS, Dürr HR, Gelderblom H, Hardes J, Hjorth L, Kreyer J, Kruseova J, Leithner A, Scobioala S, Streitbürger A, Tunn PU, Wardelmann E, Windhager R, Jürgens H, Dirksen U; GPOH-Euro-EWING99 consortium. Which Factors Are Associated with Local Control and Survival of Patients with Localized Pelvic Ewing's Sarcoma? A Retrospective Analysis of Data from the Euro-EWING99 Trial. Clin Orthop Relat Res. 2020,478(2):290-302.

We concluded that PP seems to improve outcome in a selected group of patients with pleuropulmonary involvement. However we think that at least a couple of conditions have to be fulfilled (line 224-230). We think that a good radiological response is essential, because of the importance to treat also microscopic disease. We added some sentences to discuss also the relation with the histological response:

Secondly, we think a good radiological response to chemotherapy is essential because of the importance to treat also microscopic disease, or if the size remains the same, demonstrating that the tumor has responded to chemotherapy by means of a biopsy. In pediatric bone sarcoma a good radiological response is correlated with better outcome, although a direct correlation with the histological response has not been demonstrated [27, 28]. However, it is well known that chemotherapy induced necrosis is a predictive factor for prognosis [29-31].  In 5 out of the 9 patients a poor histological response after PP, conform treatment protocols we think that radiotherapy should be added, except in patients with osteosarcoma where complete surgery could be achieved.

27 Haveman, LM, Ranft A, Vd Berg H, Smets A, Kruseova J, Ladenstein R, Brichard B, Paulussen M, Kuehne T, Juergens H, Klco-Brosius S, Dirksen U, Merks JHM. The relation of radiological tumor volume response to histological response and outcome in patients with localized Ewing Sarcoma. Cancer Med. 2019; 8(3):1086-1094.

28 Miwa S, Takeuchi A, Shirai T, Taki J, Yamamoto N, Nishida H, Hayashi K, Tanzawa Y, Kimura H, Igarashi K, Ooi A, Tsuchiya H. Prognostic value of radiological response to chemotherapy in patients with osteosarcoma. PLoS One. 2013 Jul 29;8(7):e70015.).  

29 Cotterill SJ, Ahrens S, Paulussen M, et al. Prognostic factors in Ewing's tumor of bone: analysis of 975 patients from the European intergroup Cooperative Ewing's Sarcoma Study GroupJ Clin Oncol. 2000;18(17):31083114.

30 Picci P, Rougraff BT, Bacci G, et al. Prognostic significance of histopathologic response to chemotherapy in nonmetastatic Ewing's sarcoma of the extremitiesJ Clin Oncol. 1993;11(9):17631769.

31 Sauer R, Jürgens H, Burgers JM, Dunst J, Hawlicek R, Michaelis J. Prognostic factors in the treatment of Ewing's sarcoma. The Ewing's Sarcoma Study Group of the German Society of Paediatric Oncology CESS 81Radiother Oncol 1987;10(2):101110

  • Do the authors consider that the number of 9 investigated patients is relevant for the conclusions of their study?

We included very low patient numbers, due to the rarity of patients eligible for this procedure. We agree that our conclusion is stated too firmly, however without PP all patients would have died. To emphasize that it concerns small patient numbers, we have adjusted the final conclusion:

In this very small patient group treated with a PP for primary or metastatic pleural lesions outcome was variable, however in most of the patients this extended surgical technique was generally quite well tolerated. Post-operative lung-function seemed preserved and in almost all patients quality of life was good. Because PP is the only curative option in sarcoma patients with pleural lesions, this technique can be considered as valid salvage therapy in a selected group of patients.

Also in the conclusion of the abstract we added the words: “In this small patient group…, which hopefully makes the conclusion less strict.

In this small patient group treated with a pleuropneumonectomy for primary or metastatic lesions, outcome is variable, however this extended surgical technique was generally quite well tolerated.

  • How can the authors explain if there is any possible correlation between the absence of early complications and the intensity of late complications in case of PP?

Thank you for this question. Early complications (within 4 weeks) were complications related to the surgery. The procedure went well in the majority of the patients without complications such as bleeding or infection. As described in Table 2 long-term complications (> 4 weeks after PP) resulting from, the changing anatomy or just by missing one lung. We don’t think that the incidence of long term complications is so high, neither the intensity of late complications, as this is as to be expected after extended thorax surgery. We added a recent article of Harris et al, in the discussion.

32 Harris CJ, Helenowski I, Murphy AJ, Mansfield SA, LaQuaglia MP, Heaton TE, Cavalli M, Murphy JT, Newman E, Overmen RE, Kartal TT, Cooke-Barber J, Donaher A, Malek MM, Kalsi R, Kim ES, Zobel MJ, Goodhue CJ, Naik-Mathuria BJ, Jefferson IN, Roach JP, Mata C, Piché N, Joharifard S, Sultan S, Short SS, Meyers RL, Bleicher J, Le HD, Janek K, Bütter A, Davidson J, Aldrink JH, Richards HW, Tracy ET, Commander SJ, Fialkowski EA, Troutt M, Dasgupta R, Lautz TB. Implications of Tumor Characteristics and Treatment Modality on Local Recurrence and Functional Outcomes in Children with Chest Wall Sarcoma: A Pediatric Surgical Oncology Research Collaborative Study. Ann Surg. 2020 Nov 4:10.1097

Reviewer 2 Report

Hameury et al describe a multi-institutional experience with 9 pleuro-pnuemonectomies for pediatric sarcoma.  This is an important topic and this case series represents a valuable contribution to the literature.  

A real strength of this article is the inclusion of the PFT data as well as the quality of life data.  

In addition to several more minor questions/concerns, I have one major concern.  In figure 1B, the authors show what a appears to be a solitary nodule in a patient with relapse osteosarcoma.  From the included image, I struggle to understand why the patient required such an extensive procedure.  These types of relapse are regularly and routinely managed with localized resection.  I would like to better understand the disease extent in the 3 patients with osteosarcoma who are long-term survivors.  How many nodules?  maximal nodule size?  etc.  What is the rationale for PP rather than wedge resection, lobectomy and/or more limited pleural resection?  Can you please provide more details on the preoperative extent of disease for all patients.  

Other minor questions

  1. Line 49 - the statement about RT is overly broad.  Some patients chemo- and radio-sensitive tumors (for instance, some Ewing's patients) do well without surgery
  2. Line 51 - please justify with a reference this assertion that a more limited pleurectomy is not effective
  3. Line 68 - please reword "supposed bad prognosis"
  4. Line 134 - was the indication for RT also based on the tumor histology -- some like Ewing are radiosensitive, while others like osteosarcoma are not
  5. Line 145 - you state that rib resection is not needed in pediatric patients, but it seems like all of your thoracotomy incisions included two intercostal spaces.  Please clarify.

Author Response

We thank the reviewer for the questions and the compliments. We try to answer your questions and concerns.

  • In figure 1B, the authors show what a appears to be a solitary nodule in a patient with relapse osteosarcoma.  From the included image, I struggle to understand why the patient required such an extensive procedure.  These types of relapse are regularly and routinely managed with localized resection. 

Thank you for your comment. Of course, we also treat pulmonary relapses of osteosarcoma by elective resection. In the cases presented, the problem was pleural involvement. The multidisciplinary discussions considered that the multiple pleural involvement required a complete resection of the pleura regarding response, location and number of pleural metastasis. You are quite right about the inappropriate figure (this came from patnr. 9 who appeared to have 3 pleural lesions on thoracoscopy), and is not illustrative of multiple pleural involvement. We changed it to another more appropriate example.

  • I would like to better understand the disease extent in the 3 patients with osteosarcoma who are long-term survivors.  How many nodules?  maximal nodule size?  etc. 
  • What is the rationale for PP rather than wedge resection, lobectomy and/or more limited pleural resection? 
  • Can you please provide more details on the preoperative extent of disease for all patients.  

Thank you for this important question. As mentioned above, the main reason for proposing PP was the involvement of the pleural cavity, with at least two locations, regardless of the volume of the lesions. We preferred to minimize the risk of subsequent pleural recurrence at this stage of treatment, given its dramatic prognosis.

You are right that it is necessary to have more information about the patients who survived the treatment, especially the osteosarcoma patients (Table 1, pat.nr. 1, 2, 8). Five patients were treated for osteosarcoma, and four patients are still alive, however one patients has local recurrence after PP and is treated with chemotherapy again (Table 1, pat.nr. 9). 4 out of the 5 patients had a solitary lesion at time of recurrence. In one patient (pat.nr.9, Table 1) one intrapulmonary lesion was seen at the CT scan and this was removed thorocoscopically, however at the time of surgery also 3 small pleural lesions were detected. Tumor volume at the time of recurrence differed.

In the Methods section we added a paragraph:

“Four of the five osteosarcoma patients alive were treated for a solitary lesion of the pleural lesion at time of recurrence. In one patient (pat.nr.9, Table 1) one lesion was seen at the CT  and this was removed thorocoscopically, however at the time of surgery also 3 other small pleural lesions were detected. Tumor volume at time of recurrence was different, the largest tumor measured 28x6x12cm and the smallest lesion measured 3x2.7x3cm. Before PP there was either a good radiological response, or in case the size remains the same a biopsy prior PP was taken.

In the Discussion section, we also added a sentence:

“Four of the patients alive were treated for pleuropulmonary recurrence of osteosarcoma, 3 of these patients had a solitary lesion and in one patient 3 other pleural lesions were detected at time of biopsy.”

Other minor questions

1. Line 49 - the statement about RT is overly broad.  Some patients chemo- and radio-sensitive tumors (for instance, some Ewing's patients) do well without surgery

Thank you for your remark. However we think there is some confusion about this statement. We totally agree that primitive local radiotherapy could be a choice of treatment, for example in Ewing sarcoma originating from the sacrum (ref), however, we think that primitive radiotherapy to a pleuropulmonary lesion is not a curative treatment. Hopefully this answers your remark and do you agree.

2. Line 51 - please justify with a reference this assertion that a more limited pleurectomy is not effective.

We added the references 1 and 2 to this assertion. 

3. Line 68 - please reword "supposed bad prognosis"

We changed the sentence, instead of:

“In case of recurrence of a thoracic relapse, considering the supposed bad prognosis, only lung sparing surgery with limited pleural resection is usually proposed”.

“In case of recurrence of relapse, given the poor prognosis, only surgery with limited pleural resection is often performed.”

4. Line 134 - was the indication for RT also based on the tumor histology -- some like Ewing are radiosensitive, while others like osteosarcoma are not.

Thank you for your remark. You are right that Ewing sarcoma are radiotherapy sensitive and that osteosarcoma are less sensitive. In osteosarcoma radiotherapy is only added in situations where complete surgery cannot be achieved and is recommended for inoperable sites or those that could only be operated with inadequate margins (Euramos protocol). However, in the method section we added the words ‘type of  sarcoma’:

“The indications for radiotherapy were based on positive resection margins, type of sarcoma, lymphatic emboli or poor histological response.”

In the Results section we changed line 162-163:

“Post-operative radiotherapy, either or not combined with chemotherapy, was given in order to obtain complete local control and was given because of positive resection margins, positive lymph nodes or poor histological response as decided in a multidisciplinary tumor board. This was the case in 6 patients (Table 1, pat.nr. 2-7; 3 Ewing sarcoma, 2 osteosarcoma and 1 undifferentiated sarcoma)”.

5. Line 145 - you state that rib resection is not needed in pediatric patients, but it seems like all of your thoracotomy incisions included two intercostal spaces.  Please clarify.

Although sixth rib resection is commonly performed in adults, it is not necessary in pediatric patients because of chest wall elasticity.

Thank you for your comment. In our experience, even if we had to remove a rib for oncological indications, access to the costodiaphragmatic cul de sac required an approach through the ninth intercostal space. The two intercostal space approach is preferred by the pediatric team but also by the adult team. We therefore propose to delete "Although sixth rib resection is commonly performed in adults, it is not necessary in pediatric patients because of chest wall elasticity".

Reviewer 3 Report

The authors, in this study, present a retrospective analysis to evaluate the efficacy of pleuropneumonectomy in  pediatric patients with primary or metastatic pleural sarcoma. They summarized and discussed their experience evaluating  morbidity, and with a particular focus on quality of life and long term outcome. The authors question is relevant and interesting. Further, although the sample size is very low, their  conclusions are consistent with the evidence and arguments presented.

Overall, the text is clear and easy to read.

Author Response

We thank the reviewer for the compliments and the effort made to review this article.

Round 2

Reviewer 2 Report

I am satisfied with the author's responses but insist that they remove the following phrase from the conclusion: "Because PP is the only curative option 341 in sarcoma patients with pleural lesions".  As stated in my previous comments, I do not think that sufficient evidnece has been presented to state that more limited surgical resection +/- chemo/rads can be curative in patients with pleural disease.  I certainly have patients who are long term survivors in this situation without PP.  

Author Response

Thank you. We understand your point and changed the last paragraph in the Conclusions and also the last sentence in the Simple summary -

line 5 Simple summary

“Because it gives improvement of survival in patients with pleural lesions, PP can be considered as effective salvage therapy in selected patients.”

line 299/305 Conclusions

„Although outcome is variable in patients treated with a pleuropneumonectomy for primary or metastatic pleural lesions, this extended surgical technique is generally quite well tolerated. Post-operative lung-function seems well preserved and in almost all patients quality of life is good. To our knowledge no other treatment demonstrated such improvement of survival, even if it has to be confirmed with largest studies. Therefore, PP seems to be an alternative to palliative treatment in a selected group of young patients with pleural sarcoma.”